# *FGFR* Fusions as an Acquired Resistance Mechanism Following Treatment with Epidermal Growth Factor Receptor Tyrosine Kinase Inhibitors (EGFR TKIs) and a Suggested Novel Target in Advanced Non-Small Cell Lung Cancer (aNSCLC)

**DOI:** 10.3390/jcm11092475

**Published:** 2022-04-28

**Authors:** Ari Raphael, Elizabeth Dudnik, Dov Hershkovitz, Suyog Jain, Steve Olsen, Lior Soussan-Gutman, Taly Ben-Shitrit, Addie Dvir, Hovav Nechushtan, Nir Peled, Amir Onn, Abed Agbarya

**Affiliations:** 1Department of Oncology, Tel-Aviv Sourasky Medical Center, Tel-Aviv 6423906, Israel; arir@tlvmc.gov.il; 2Sackler Faculty of Medicine, Tel-Aviv University, P.O. Box 39040, Ramat Aviv, Tel-Aviv 6997801, Israel; dovh@tlvmc.gov.il; 3Thoracic Oncology Service, Davidoff Cancer Center, Rabin Medical Center, Beilinson Campus, Petah Tikva 4941492, Israel; 4Department of Pathology, Tel-Aviv Sourasky Medical Center, Tel-Aviv 6423906, Israel; 5Guardant Health Asia Middle East and Africa, Singapore 138543, Singapore; sjain@guardantamea.com (S.J.); solsen@guardantamea.com (S.O.); 6Rhenium Oncotest Ltd., Ha-Satat St. 20, Modi’in-Maccabim-Re’ut 7177704, Israel; liorsg@rhenium.co.il (L.S.-G.); taly@rhenium.co.il (T.B.-S.); addie@rhenium.co.il (A.D.); 7Department of Oncology, Hadassah Medical Organization and Faculty of Medicine, Hebrew University of Jerusalem, P.O. Box 12000, Jerusalem 9112102, Israel; hovavnech@hadassah.org.il; 8Department of Oncology, Shaare Zedek Medical Center, Jerusalem 9103102, Israel; peled.nir@gmail.com; 9Pulmonology Institute, Sheba Medical Center, Tel HaShomer, Ramat Gan 5262000, Israel; amir.onn@sheba.health.gov.il; 10Department of Oncology, Bnai Zion Medical Center, Haifa 3339419, Israel; abed.agbarya@b-zion.org.il

**Keywords:** *FGFR*, *FGFR* fusion, acquired resistance to EGFR TKIs, lung cancer, liquid biopsy, ctDNA

## Abstract

Background. *FGFR1/2/3* fusions have been reported infrequently in aNSCLC, including as a rare, acquired resistance mechanism following treatment with EGFR TKIs. Data regarding their prevalence and therapeutic implications are limited. Methods. The Guardant Health (GH) electronic database (ED) was evaluated for cases of aNSCLC and *FGFR2/3* fusions; *FGFR2/3* fusion prevalence with and without a co-existing *EGFR* mutation was assessed. The ED of Tel-Aviv Sourasky Medical Center (TASMC, June 2020–June 2021) was evaluated for cases of aNSCLC and de novo *FGFR1/2/3* fusions. Patients with *EGFR* mutant aNSCLC progressing on EGFR TKIs and developing an *FGFR1/2/3* fusion were selected from the ED of Davidoff Cancer Center (DCC) and Oncology Department, Bnei-Zion hospital (BZ) (April 2014–April 2021). Clinicopathological characteristics, systemic therapies, and outcomes were assessed. Results. In the GH ED (*n* = 57,445), the prevalence of *FGFR2* and *FGFR3* fusions were 0.02% and 0.26%, respectively. *FGFR3-TACC3* fusion predominated (91.5%). In 23.8% of cases, *FGFR2/3* fusions co-existed with *EGFR* sensitizing mutations (exon 19 del, 64.1%; L858R, 33.3%, L861Q, 2.6%). Among samples with concurrent *FGFR* fusions and *EGFR* sensitizing mutations, 41.0% also included *EGFR* resistant mutations. In TASMC (*n* = 161), 1 case of de novo *FGFR3-TACC3* fusion was detected (prevalence, 0.62%). Of three patients from DCC and BZ with *FGFR3-TACC3* fusions following progression on EGFR TKIs, two received EGFR TKI plus erdafitinib, an FGFR TKI, with clinical benefit duration of 13.0 and 6.0 months, respectively. Conclusions. Over 23% of *FGFR2/3* fusions in aNSCLC may be associated with acquired resistance following treatment with EGFR TKIs. In this clinical scenario, a combination of EGFR TKIs and FGFR TKIs represents a promising treatment strategy.

## 1. Introduction

Epidermal growth factor receptor tyrosine kinase inhibitors (EGFR TKIs) represent an extremely effective therapy option for a distinct yet substantial subset of patients with advanced non-small lung cancer (aNSCLC) harboring *EGFR* activating mutations. Recently, they have also demonstrated a benefit in the adjuvant setting [1]. Unfortunately, the development of acquired resistance during EGFR TKI treatment is inevitable [2]. The mechanisms of acquired resistance to EGFR TKIs are complex and heterogeneous and often involve the co-activation of several molecular pathways. The most common resistant mechanisms involve the development of secondary resistant mutations in the *EGFR* kinase domain, such as *EGFR* T790M in approximately half of the tumors following treatment with first- or second-generation EGFR TKIs or *EGFR* C797S following therapy with osimertinib, which occurs in 10–26% of cases [3,4]. Among the “oncogene kinase switch”-resistant mechanisms are tyrosine-protein kinase Met (*cMET*)-amplification, which occurs in 5–50% of cases, phosphatidylinositol-4,5-bisphosphate 3-kinase (*PIK3CA*) mutations (in 4–11% of cases), Kirsten rat sarcoma viral oncogene homolog (*KRAS*) mutations (in 2–8% of cases), v-raf murine sarcoma viral oncogene homolog B1 (*BRAF*) mutations (5–8% of cases) [5,6,7,8,9], receptor tyrosine kinase (RTK) fusions (e.g., rearranged during transfection (*RET*)—coiled-coil domain-containing protein 6 *(CCDC6*), anaplastic lymphoma kinase (*ALK*)—echinoderm microtubule-associated protein-like 4 (*EML4*), and fibroblast growth factor receptor 3 (*FGFR3*)—transforming acidic coiled-coil–containing protein 3 (*TACC3*) fusions (which occur in about 1% of cases each) [10,11]. Ou et al. reported five cases of *FGFR3-TACC3* fusion co-existing with an activating *EGFR* mutation in the setting of acquired resistance to EGFR TKIs in a large database of aNSCLC tissue samples analyzed by a comprehensive next-generation sequencing platform [12]. Acquisition of tumor tissue is more challenging in the clinical scenario of acquired resistance following EGFR TKIs treatment, given the spatial and temporal heterogeneity of resistance mechanisms and the need to subject a patient to additional invasive procedure(s) to acquire a timely tissue specimen reflective of the genomic landscape at the time of progression. This is the clinical setup in which liquid biopsy may be of value, eliminating the need for repeated tissue biopsy and allowing for the detection of molecular resistance mechanisms.

Based on the preclinical data, FGFR tyrosine kinase inhibition (FGFR TKIs) leads to G2/M cell cycle blockade and cell proliferation arrest [13]. Two different FGFR TKIs were recently approved by the United States Food and Drug Administration Agency (US FDA) for clinical use in tumors harboring *FGFR* alterations: pemigatinib, for the treatment of advanced intrahepatic cholangiocarcinoma, and erdafitinib, for the treatment of refractory urothelial cancers [13,14]. The clinical data regarding FGFR TKIs as a treatment for *FGFR*-rearranged aNSCLC is limited. Twenty-four patients with aNSCLC (eight of those with an *FGFR* fusion) were included in a phase 1 trial of erdafitinib in patients with advanced or refractory solid tumors. The objective response rate with erdafitinib in 21 evaluable aNSCLC patients was 5% [15]. Additionally, one patient with an *EGFR* mutant aNSCLC that developed an *FGFR3-TACC3* fusion following treatment with osimertinib has been reported. The patient received a combination of erdafitinib and osimertinib; the treatment was well tolerated, and a partial response was achieved [16].

We aimed to further explore the prevalence and therapeutic implications of *FGFR* fusions in aNSCLC, mainly focusing on the acquired resistance setting following treatment with EGFR TKIs.

## 2. Materials and Methods

### 2.1. Guardant Health (GH) Electronic Database (ED)

*FGFR2/3* fusion frequency and *EGFR* mutation co-occurrence in plasma samples were calculated from the ED of Guardant360 CDx and Guardant360 LDT version 2.11 (genomic results from 57445 patients). Guardant360 is not designed to detect *FGFR1* fusions. The database included results of plasma samples with successfully extracted circulating tumor deoxyribonucleic acid (ctDNA) from individuals with aNSCLC undergoing testing as part of their routine care. For the purposes of this analysis, all samples were de-identified, and corresponding clinical treatment history was not available.

Guardant360 is a Clinical Laboratory Improvement Amendments (CLIA)-certified, College of American Pathologists (CAP)-accredited, New York State Department of Health–approved laboratory, and FDA-approved assay [17,18]. The assay detects single-nucleotide variants (SNV), insertions and deletions (indels), fusions, and copy number variations in 74 genes with a reportable range of ≥0.04%, ≥0.02%, ≥0.04%, and ≥2.12 copies, respectively, as well as microsatellite instability. ctDNA was extracted from whole blood collected in 10-mL Streck tubes. After double ultracentrifugation, 5–30 ng of ctDNA was isolated from plasma for digital sequencing. After the isolation of ctDNA by hybrid capture, the assay was performed using molecular barcoding and proprietary bioinformatics algorithms with massively parallel sequencing on an Illumina Hi-Seq 2500 platform in a single laboratory (Guardant Health; Redwood City, CA, USA) [17,18].

### 2.2. Tel-Aviv Sourasky Medical Center (TASMC) ED

The frequency of de novo *FGFR1/2/3* fusion and *EGFR* mutation co-occurrence in the tumor tissue specimens were determined using Tel-Aviv Sourasky Medical Center (TASMC) clinicopathological ED. TASMC represents a referral center for upfront tumor molecular testing. The mentioned database comprised of tumor samples from 161 aNSCLC patients collected in June 2020–June 2021. NGS was done using Archer^®^ VariantPlex^®^ and FusionPlex^®^ Comprehensive Thyroid and Lung kit. Briefly, total nucleic acid (DNA and ribonucleic acid [RNA]) was extracted using the QIAamp DNA formalin-fixed paraffin-embedded (FFPE) Tissue kit (Qiagen Inc., Hilden, Germany). Libraries were generated using the Archer^®^ VariantPlex^®^ Thyroid and Lung kit and the Archer FusionPlex^®^ Lung panels (Archer, Boulder, CO, USA). Libraries were run on MiniSeq platform. Binary alignment map (BAM) files were uploaded into the Archer data analysis pipeline (Archer™ analysis software version).

### 2.3. Patients with FGFR1/2/3 Fusion as an Acquired Resistance Mechanism Following Treatment with EGFR TKIs

Patients with an *EGFR* mutant aNSCLC progressing on EGFR TKIs and developing an *FGFR1/2/3* fusion (detected either in the ctDNA or in the tumor tissue specimen) were selected from the ED of Davidoff Cancer Center (DCC) and the Oncology Department of Bnei-Zion hospital (BZ) (April 2014–April 2021, *n* = 3). Clinicopathological patients’ characteristics and systemic therapy outcomes were assessed. The response assessment was done either using computer tomography (CT) or fluorodeoxyglucose positron emission tomography/computer tomography (FDG-PET/CT) every 8–12 weeks.

### 2.4. Ethical Aspects

Institutional review board approval was received before study initiation in TASMC ED. No patient-identifying data was included in the central data collection.

## 3. Results

### 3.1. FGFR2/3 Fusion Frequency in GH Electronic Database

This analysis was done on the validated dataset in GH ED and comprised genomic data from 57,445 patients with aNSCLC. A total of 164 (0.28% of all aNSCLC cases) carried either *FGFR2 or FGFR3* fusion; the prevalence of *FGFR2* and *FGFR3* fusions were 0.02% and 0.26%, respectively (Figure 1). Approximately 50% of tumors with *FGFR* rearrangements were adenocarcinoma, 38% were squamous cell carcinoma, and the remaining cases were of either mixed or unknown histology (Table 1). Amongst all *FGFR* fusion subtypes, *FGFR3-TACC3* fusions were the most abundant (91.46% of cases); other fusion partners were *TACC2*, Bicaudal C homolog 1 (*BICC1)*, shootin-1 (*SHTN1*, also called *KIAA1598)*, *CCDC6*, solute carrier family 45 member 3 (*SLC45A3*), and Wolf-Hirschhorn syndrome candidate gene-1 (*WHSC1)* (Figure 2A, Table 1). The median *FGFR* mutant allele frequency (MAF) was 0.5%.

Out of all the 164 patients with *FGFR2/3* fusions, 39 (23.78%) patients (0.07% of all aNSCLC cases) had concurrent *EGFR* sensitizing mutations (Figure 1 and Figure 2B). In samples with *FGFR2/3* fusions with co-existing *EGFR* mutations, subtypes of the *EGFR* sensitizing mutation were as follows: exon 19 del—64.1%, L8585R—33.3%, and L861Q—2.6% (Figure 2B, Table 2). Among the 39 patients with *FGFR2/3* fusions and concurrent *EGFR* sensitizing mutations, co-existing *EGFR* resistance mutations (T790M and/or C797X) were observed in 16 (41.02%) patients (0.03% of all aNSCLC cases) (Figure 1). Those *EGFR* resistant mutation types were as follows: T790M—20.5%, C797X—2.6%, and both T790M and C797X—17.9% (Table 2).

### 3.2. FGFR1/2/3 Fusion Frequency in TASMC Electronic Database

The TASMC ED included 161 cases with aNSCLC, among which there was one case of de novo *FGFR3-TACC3* fusion (prevalence—0.62%). This tumor did not harbor a co-existing *EGFR* mutation.

### 3.3. Case-Series of Patients with FGFR3-TACC3 Fusion as an Acquired Resistance Mechanism Following Treatment with EGFR TKIs

Three patients with *EGFR* mutant aNSCLC progressing on EGFR TKIs and developing an *FGFR3-TACC3* fusion were identified in the ED of DCC and BZ. In all the three cases, *FGFR3-TACC3* fusion was detected by Guardant360 performed at radiological progression on osimertinib. Clinicopathological patient characteristics are summarized in Table 3.

### 3.4. Clinical Case #1

The first patient was a 59-year-old never-smoking female diagnosed with lung adenocarcinoma harboring an *EGFR* L858R substitution on exon 21 (verified by polymerase chain reaction [PCR] in the tumor specimen) with bilateral lung metastases, liver metastases, and a left-sided malignant pleural effusion. Gefitinib was initiated, and a partial response was achieved. Seven months later, upon disease progression, droplet digital PCR (ddPCR) of the plasma was performed and revealed the presence of an *EGFR* T790M mutation. Gefitinib was replaced by osimertinib with a partial response lasting 13 months. Upon radiological disease progression in the lungs and pleura, treatment was changed to platinum and pemetrexed, with disease stabilization for an additional 6 months. Eventually, following a disease progression, blood was collected for Guardant360 testing, which revealed an *EGFR* L858R mutation (MAF 33.4%), *PIK3CA* E545K mutation (MAF 47.5%), *FGFR3-TACC3* fusion (MAF 0.3%), cyclin D1 (*CCND1)* amplification, cyclin-dependent kinase 4 (*CDK4*) amplification, *KRAS* amplification, and Myelocytomatosis viral oncogene homolog (*MYC*) amplification. The patient did not receive any subsequent systemic therapy and died of aNSCLC 3 months thereafter.

### 3.5. Clinical Case #2

The second patient is an 84-year-old never-smoking male who was diagnosed with advanced-stage lung adenocarcinoma, a malignant right-sided pleural effusion, and bone metastases. PCR of the pleural fluid cell-block revealed the presence of an *EGFR* exon 19 deletion (E746_A750del). Osimertinib 80 mg once daily was initiated, and a partial response was achieved. Eleven months after treatment initiation, the disease progressed with the development of new mediastinal lymphadenopathy and recurrent right-sided pleural effusion necessitating the insertion of a drain. At that point, Guardant360 testing was performed and, in addition to detecting the original sensitizing mutation *EGFR* E746_A750del (MAF 1.3%), identified a novel *FGFR3-TACC3* fusion (MAF 0.04%) along with tumor protein P53 (*TP53*) Y163C (MAF 0.4%). Erdafitinib 8 mg was added to osimertinib 80 mg daily, resulting in mediastinal lymph node shrinkage, stabilization of the lung metastasis and pleural effusion, and calcification of lytic bone metastases. All metastatic sites showed a reduction in fluorodeoxyglucose (FDG)-avidity in the FDG-PET/CT (Figure 3). The treatment was well-tolerated; no laboratory abnormalities were seen. The patient continued with combination TKI treatment with disease stabilization lasting for 6 months since treatment initiation.

### 3.6. Clinical Case #3

The third patient is a 63-year-old never-smoking female who was diagnosed with advanced-stage lung adenocarcinoma harboring an *EGFR* L747_A750delinsP mutation (exon 19 deletion). She was initially treated with gefitinib, with partial response lasting 52 months. Upon radiological disease progression, treatment was changed to osimertinib following the detection of an *EGFR* T790M mutation by ddPCR. Eleven months later, disease spread to the peritoneum (omental cake) and was confirmed by peritoneal biopsy. NGS (Tempus xT) was performed on the tissue specimen and revealed an *EGFR* L747_A750delinsP (MAF 14.4%), *EGFR* C797S (MAF 3.6%), and *PIK3CA* V344G (MAF 15.9%). Guardant360 performed on plasma collected at the same time revealed the presence of an *FGFR3-TACC3* fusion (MAF 0.07%) in addition to the previously described molecular alterations (*EGFR* L747_A750delinsP, *PIK3CA* V344G). Treatment with erdafitinb 9 mg and gefitinb 250 mg daily was initiated with clinical improvement and radiological disease stabilization. Four months later, a solitary brain metastasis was detected. Stereotactic radiosurgery was conducted, and combined systemic treatment was continued. Due to isolated asymptomatic grade 3 alanine transaminase (ALT) elevation, the erdafitinib dose was reduced to 4 mg daily. Nine months after the initiation of the combined TKI treatment, the disease progressed systemically and intracranially, and systemic treatment was changed to carboplatin with pemetrexed.

## 4. Discussion

Analysis of FGFR2/3 fusions in aNSCLC patients in the GH electronic database, the largest analysis yet reported on the subject, revealed that *FGFR2/3* fusions co-exist with *EGFR* sensitizing mutations in 24% of cases, and in 41% of those cases, concurrent *EGFR* resistant mutations are also seen. This observation suggests that *FGFR2/3* fusions may represent a rare but clinically important acquired molecular resistance mechanism following treatment with EGFR TKIs in *EGFR* mutant aNSCLC. Moreover, in the presented clinical series, two out of three patients with an *FGFR3-TACC3* fusion, as a confirmed acquired molecular resistance mechanism to EGFR TKIs, derived clinical benefit from adding an FGFR TKI, erdafitinib, to EGFR TKI.

According to the GH electronic database, the prevalence of *FGFR2/3* fusions in liquid biopsy specimens from patients with aNSCLC was 0.28%, whereas the analysis of TASMC electronic database demonstrated *FGFR* fusions prevalence in tumor tissue specimens of 0.62%. Both results are in line with previous reports on *FGFR2/3* fusions prevalence ranging between 0.2% and 1.3% when assessed by comprehensive molecular profiling. For instance, Qin et al. [19] reported *FGFR* fusions retaining the kinase domain in 0.2% (52 of 26.054 NSCLC cases), including 37 *FGFR3-TACC3* fusions, two *FGFR2* fusions, one *FGFR1* fusion (all previously reported), and 12 novel *FGFR1*, *FGFR2*, *FGFR3*, and *FGFR4* fusions. Wang et al. reported a *FGFR* fusion prevalence of 1.3% in a cohort of 1.328 patients with NSCLC, amd the majority of cases (15 of 17) were *FGFR3-TACC3* fusions [20]. Additionally, Capelletti et al. reported a *FGFR3-TACC3* fusions prevalence of 0.5% in a cohort of 576 patients with lung adenocarcinoma [21]. The differences in *FGFR* rearrangements prevalence may result from the techniques used for the assessment of fusions in different assays. Specifically, the technical limitations of DNA-based NGS for detection of fusions may lead to a slightly lower prevalence of *FGFR* fusions compared to assays that assess for gene rearrangements using RNA or protein. On the other hand, tissue-based NGS databases are mainly composed of samples from treatment-naive patients, as opposed to plasma-based NGS databases which are enriched for patients who have had multiple lines of systemic treatment, hence increasing the probability of finding *FGFR* fusions representing an acquired resistance mechanism.

Amongst all the *FGFR* fusion subtypes, *FGFR3-TACC3* predominated; this subtype represented 91% of all *FGFR* fusions in the GH electronic database. This observation is in line with the previously reported data regarding the frequency of *FGFR3-TACC3* fusions in *FGFR*-rearranged NSCLC (77–88%) [19,20].

In the GH electronic database, *FGFR2/3* fusions were more frequently seen in lung adenocarcinomas as compared to other histological lung cancer subtypes. The previously reported data, on the other hand, points to the two most common NSCLC histological subtypes (adenocarcinoma and squamous cell carcinoma) as the main source of *FGFR* fusions [19,20]. Additionally, in the GH database, about one-quarter of patients with *FGFR* fusions had co-existing *EGFR* sensitizing mutations. Nearly half of those also harbored one of the *EGFR* resistant mutations, suggesting possible tumor progression on 1st or 2nd-generation EGFR TKI before sample acquisition. The higher frequency of adenocarcinoma-associated *FGFR* fusions and the higher frequency of co-existing *EGFR* sensitizing and *EGFR* resistant mutations in the GH electronic database may both reflect the higher proportion of cases with acquired resistance to targeted treatments included in the GH database, which, in turn, reflects real-world referral patterns for liquid biopsy testing after disease progression on prior therapies.

The acquisition of *FGFR3-TACC3* fusion has been described as a molecular resistance mechanism following treatment with EGFR TKIs in *EGFR* mutant aNSCLC [12,22]. We observed that 0.06% of aNSCLC samples in the GH database had *FGFR3-TACC3* fusion with a concurrent *EGFR* mutation, similar to the results reported by Ou et al. (0.03%; 5 of 17,319 aNSCLC cases) using either tumor tissue or plasma as source material [12], but somewhat greater than those reported by Quin et al. (0.007%) using tumor tissue only [19].

It is difficult to estimate the exact prevalence of acquired *FGFR* fusion following treatment with EGFR TKIs in *EGFR* mutant aNSCLC because most large molecular databases do not include testing and/or treatment histories. In the absence of such information in the GH database, we assumed that in cases with both *EGFR* sensitizing mutation and *FGFR* fusion, the former represented the original oncogenic driver. The reported prevalence is based on the following assumptions: first, the co-existing *EGFR* resistant mutations reflect the underlying molecular resistant mechanism, and then, all the tumors developing an *FGFR* rearrangement following the progression on EGFR TKIs retain the original *EGFR* mutation, which may or may not be true. We also acknowledge that most cases of *FGFR* fusions in GH ED occurred without a concomitant *EGFR* mutation, and therefore, in most cases, *FGFR* fusion might represent a driver by itself.

The clinical data on FGFR TKIs as a treatment of de novo *FGFR*-aberrated aNSCLC, as opposed to intrahepatic cholangiocarcinoma and urothelial carcinoma, is limited to 24 cases treated with erdafitinib in a phase 1 trial [15]. The objective response rate with erdafitinib in 21 evaluable aNSCLC patients was 5%, efficacy in 8 patients with de novo *FGFR*-rearrangements has not been reported [15]. In the clinical scenario of acquired *FGFR-* fusions following treatment with osimertinib in *EGFR* mutant aNSCLC, one patient achieved a partial response with the combination of erdafitinib and osimertinib has been previously reported [16], and our case-series adds to the reported data.

This study highlights the important role of ctDNA testing as a sensitive and non-invasive approach for the detection of emerging and potentially actionable genomic alterations that may develop in aNSCLC after treatment with targeted therapies. Recently, the International Association for the Study of Lung Cancer (IASLC) updated its liquid biopsy guideline and recommended the use of ctDNA as the first method (“plasma first”) to discover molecular resistance mechanisms—due to the simplicity and high patient advocacy [23]. Such testing may help to identify actionable alterations, such as *FGFR3* fusions that inform effective targeted therapeutic options.

## 5. Conclusions

In conclusion, although *FGFR* fusions represent a rare molecular event in the setting of acquired resistance to EGFR TKIs, its finding enables an emerging treatment strategy of combining EGFR TKIs with FGFR TKIs. Such combinations warrant a prospective evaluation.

## Figures and Tables

**Figure 1 jcm-11-02475-f001:**
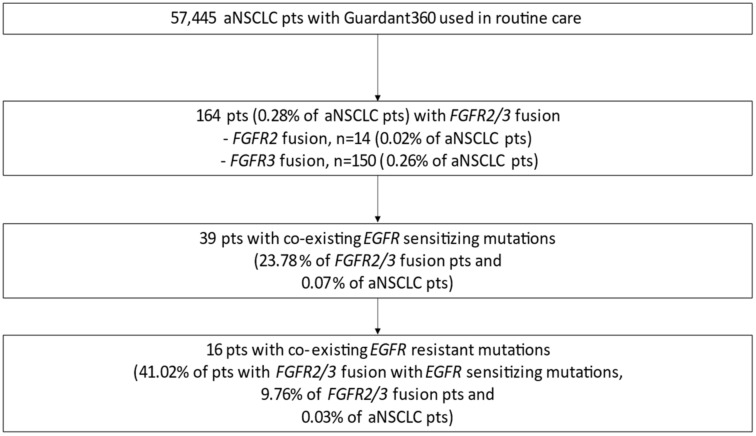
Prevalence of FGFR2/3 fusions and co-existing EGFR sensitizing and resistant mutations in aNSCLC in the GH electronic database. Abbreviations: aNSCLC—advanced non-small cell lung cancer; EGFR—epidermal growth factor receptor; FGFR—fibroblast growth factor receptor; GH—Guardant Health; pts—patients.

**Figure 2 jcm-11-02475-f002:**
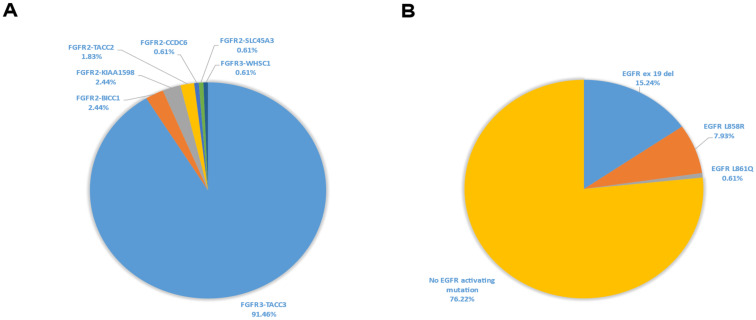
*FGFR2/3* fusion subtypes distribution (**A**) and prevalence of co-existing *EGFR* sensitizing mutations (**B**) in aNSCLC in the GH electronic database. Abbreviations: aNSCLC—advanced non-small cell lung cancer; *EGFR*—epidermal growth factor receptor; *FGFR*—fibroblast growth factor receptor; GH—Guardant Health.

**Figure 3 jcm-11-02475-f003:**
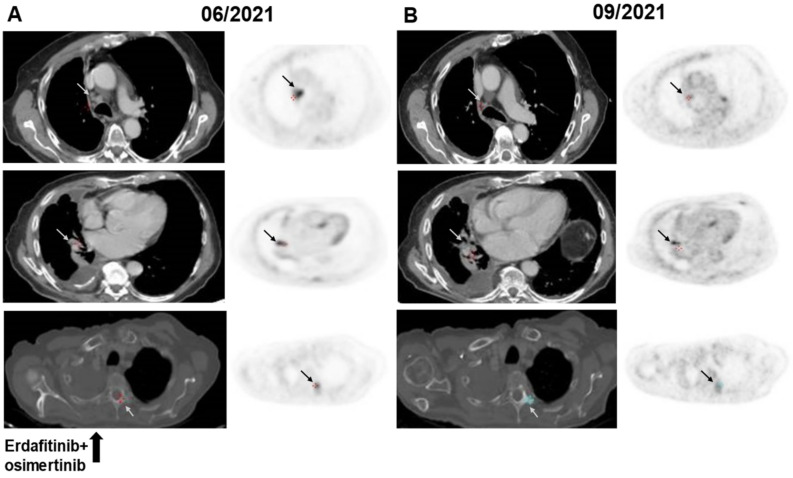
FDG-PET/CT images before (**A**) and during (**B**) therapy with osimertinib + erdafitinib in a patient with an *EGFR* mutated aNSCLC and *FGFR3-TACC3* fusion following progression on osimertinib. Shrinkage of a retro-caval lymph node with a reduction in FDG-avidity, stable lung metastasis with a reduction in FDG-avidity, calcification of a D3 lytic bone metastasis with a reduction in FDG-avidity (gray and black arrows). Abbreviations: aNSCLC—advanced non-small cell lung cancer; *EGFR*—epidermal growth factor receptor; FDG—*fluorodeoxyglucose*; *FGFR3-TACC3*—fibroblast growth factor receptor 3-transforming acidic coiled-coil-containing protein 3; PET/CT—*positron emission tomography*/computer tomography.

**Table 1 jcm-11-02475-t001:** *FGFR2/3* fusion subtype prevalence and distribution across different histological non-small cell lung cancer subtypes in the Guardant Health electronic database.

*FGFR* Fusion Type	Large-Cell Carcinoma	ADC	Adeno-Squamous Carcinoma	SCC	NSCLC NOS Carcinoma	Total
*FGFR2-BICC1*		4				4
*FGFR2-CCDC6*		1				1
*FGFR2-KIAA1598*		3		1		4
*FGFR2-SLC45A3*					1	1
*FGFR2-TACC2*		2			1	3
*FGFR3-TACC3*	4	71	1	61	13	150
*FGFR3-WHSC1*					1	1
Total	4	81	1	62	16	164

Abbreviations: ADC—adenocarcinoma; *FGFR*—fibroblast growth factor receptor; NSCLC NOS—non-small cell lung cancer non otherwise specified carcinoma; SCC—squamous cell carcinoma.

**Table 2 jcm-11-02475-t002:** *EGFR* mutation types co-occurring with *FGFR2/3* fusions in the Guardant Health electronic database.

*FGFR* Fusion Type	*EGFR* Sensitizing Mutation Type				*EGFR* Resistance Mutation Type			
	Exon 19 Deletion, *n* (%)	L858R, *n* (%)	L861Q, *n* (%)	Total, *n* (%)	T790M, *n* (%)	C797X, *n* (%)	T790M and C797X, *n* (%)	None, *n* (%)
*FGFR2-CCDC6*	1 (2.6)	0 (0)	0 (0)	1 (2.6)	1 (2.6)	0 (0)	0 (0)	0 (0)
*FGFR2-KIAA1598*	1 (2.6)	1 (2.6)	0 (0)	2 (5.1)	1 (2.6)	0 (0)	0 (0)	1 (2.6)
*FGFR2-TACC2*	0 (0)	1 (2.6)	0 (0)	1 (2.6)	1 (2.6)	0 (0)	0 (0)	0 (0)
*FGFR3-TACC3*	23 (58.9)	11 (28.2)	1 (2.6)	35 (89.7)	5 (12.8)	1 (2.6)	7 (17.9)	22 (56.4)
Total	25 (64.1)	13 (33.3)	1 (2.6)	39 (100)	8 (20.5)	1 (2.6)	7 (17.9)	23 (59.0)

Abbreviations: *EGFR*—epidermal growth factor receptor; *FGFR*—fibroblast growth factor receptor.

**Table 3 jcm-11-02475-t003:** Demographic and clinico-pathological characteristics of patients with *EGFR* mutant aNSCLC progressing on EGFR TKIs and developing an *FGFR3-TACC3* fusion. All *FGFR3-TACC3* fusions were detected by Guardant 360.

Case Number	Sex	Age, Years	Tumor Histology	Smoking History	*EGFR* Mutation Subtype	Treatment History before *FGFR3-TACC3* Fusion Diagnosis: Agent (PFS, mo)	*FGFR3-TACC3* Fusion MAF, %	Concurrent Alterations, MAF, %
#1	F	59	ADC	Never- smoker	L858R	Gefitinib (7 mo), osimertinib (13 mo), carboplatin/pemetrexed (6 mo)	0.3	*EGFR* L858R, 33.4, *PIK3CA* E545K, 47.5, *CCND1* amplification, *CDK4* amplification, *KRAS* amplification, *MYC* amplification
#2	M	84	ADC	Never- smoker	E746_A750del	Osimertinib (11 mo)	0.04	*EGFR* E746_A750del, 1.3, *TP53* Y163C, 0.4
#3	F	63	ADC	Never- smoker	L747_A750delinsP	Gefitinib (52 mo), osimertinib (14 mo)	0.07	Gardant360: *EGFR* L747_A750delinsP, 0.5, *PIK3CA* V344G, 1.3 Tempus xT: *EGFR* L747_A750delinsP, 14.4, *EGFR* p. C797S, 3.6, *PIK3CA* V344G, 15.9

Abbreviations: ADC—adenocarcinoma; *EGFR*—epidermal growth factor receptor; F—female; *FGFR*—fibroblast growth factor receptor; M—male; MAF—mutant allele frequency; mo—months; PFS—progression-free survival.

## Data Availability

According to the GH policy, the access to the database cannot be provided.

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
