# Peer review of "FGFR Fusions as an Acquired Resistance Mechanism Following Treatment with Epidermal Growth Factor Receptor Tyrosine Kinase Inhibitors (EGFR TKIs) and a Suggested Novel Target in Advanced Non-Small Cell Lung Cancer (aNSCLC)"

_jcm, 2022, doi:10.3390/jcm11092475_

Round 1
Reviewer 1 Report
In this report titled: “FGFR fusions as an acquired resistance mechanism following treatment with epidermal growth factor receptor tyrosine kinase inhibitors and a suggested novel target in advanced non-small cell lung cancer”, Raphael et al. present an observational study in different and valuable advanced NSCLC patient databases. As others did before, they find a relatively low prevalence of FGFR fusions, and some of them occurring de novo upon anti-EGFR TKI treatment. For this, they take advantage of the Guardant Health electronic database, with information of liquid biopsy to detect FGFR fusions, together with other local databases. They also show real results in this clinical scenario in a small set of patients (n=2), in which patients developing resistance to anti-EGFR treatment, get considerably clinical benefit under clinically available anti-FGFR therapy. The paper is easy to read and clear, in general terms. While the concept is not really novel (authors cite some similar reports very adequately), this report adds information with potential implication to precision oncology of aNSCLC. Out of the numerous patients under anti-EGFR therapies, some will eventually acquire resistance through FGFR related mechanisms. The article points to two relevant aspects of the clinical scenario: the potential of liquid biopsy to monitor known resistance mechanisms and the possible tailored therapy for arising FGFR fusions. The percentage of patients with these fusions might seem small, but this is the basis of precision oncology.
Here I add some comments:
Abstract
- It would be good to mention the sample size for each cohort in the abstract (as they do for the last one n=3).
- It would also help to non-expert readers to state that erdafitinib is an anti-FGFR therapy.
Introduction
- Although this is well elaborated in the discussion section, this reviewer would emphasize the relevance of liquid biopsy, the advantages, significance, limitations, and value in resistance detection, in tumors that are rarely re-biopsied.
Material and methods
- Since there is data from sequencing of cfDNA and tumor tissue material, this should be clarified before in the text for all ED because the interpretation of the results depends on that.
- In the in-house TASMC cohort, authors provide some details on the NGS kit and genetic material employed. If possible, they should provide more details on the NGS protocol. Does the FusionFlex kit use DNA or RNA material? How was the quality assessed? These results might be important to show as supp data, because the genetic material from FFPE might compromise the results.
- Some information on the PET-CT scans (which is in the discussion) should be included as material and methods.
Results
- The first two sections might be fused in one as ‘FGFR2/3 fusion frequency in electronic databases’. In the different paragraphs with the information from GH and TASMC EDs, difference on the origin of the sample (plasma and tumor tissue) should be clearly stated.
- Text in Figure 2 pie charts should be larger to be readable.
- This reviewer thinks that supp tables could be presented as main, and not supplementary.
- In Table s1, authors could add percentages to the number of cases. Although some percentages are displayed in the pie chart, this would more clearly associate these two interpretations of the numbers.
- I would recommend using more widely used abbreviations, such as LUAD or ADC (or ADC-SCC), instead of adenoca.
Discussion
- Discussion is very complete and clarifying.
- Major: Regarding the statement, “We also acknowledge that most cases of FGFR fusions in GH ED occurred without a concomitant EGFR mutation, and
- therefore, in the majority of cases, FGFR fusion represents a driver by itself.”, Would you have information on other driver co-mutations in these patients? This would be relevant information to add even in one more figure if possible.
- Little discussion on the occurrence of these fusions in patients not subsidiary of other approved therapies, and the occurrence of these fusions in other tumor types, might expand the impact of these results beyond lung tumors.
Author Response
Dr. Emmanuel Andrès
Journal of Clinical Medicine
Editor-in-Chief
April 20, 2022
Re: “FGFR fusions as an acquired resistance mechanism following treatment with epidermal growth factor receptor tyrosine kinase inhibitors (EGFR TKIs) and a suggested novel target in advanced non-small cell lung cancer (aNSCLC)”
Dear Dr. Emmanuel Andrès,
Dear Reviewer,
On behalf of my co-authors, I would like to thank you for considering our revised manuscript for publication in Journal of Clinical Medicine. We are grateful for the helpful comments of the reviewers and for the editorial work. Considerable corrections have been made according to reviewers' comments – please, find below the list of points which were addressed.
Reviewer #1:
In this report titled: “FGFR fusions as an acquired resistance mechanism following treatment with epidermal growth factor receptor tyrosine kinase inhibitors and a suggested novel target in advanced non-small cell lung cancer”, Raphael et al. present an observational study in different and valuable advanced NSCLC patient databases. As others did before, they find a relatively low prevalence of FGFR fusions, and some of them occurring de novo upon anti-EGFR TKI treatment. For this, they take advantage of the Guardant Health electronic database, with information of liquid biopsy to detect FGFR fusions, together with other local databases. They also show real results in this clinical scenario in a small set of patients (n=2), in which patients developing resistance to anti-EGFR treatment, get considerably clinical benefit under clinically available anti-FGFR therapy. The paper is easy to read and clear, in general terms. While the concept is not really novel (authors cite some similar reports very adequately), this report adds information with potential implication to precision oncology of aNSCLC. Out of the numerous patients under anti-EGFR therapies, some will eventually acquire resistance through FGFR related mechanisms. The article points to two relevant aspects of the clinical scenario: the potential of liquid biopsy to monitor known resistance mechanisms and the possible tailored therapy for arising FGFR fusions. The percentage of patients with these fusions might seem small, but this is the basis of precision oncology.
Here I add some comments:
Abstract
- It would be good to mention the sample size for each cohort in the abstract (as they do for the last one n=3).
Response: Thank you for your valuable comment! Was added according to your suggestion (p. 2).
- It would also help to non-expert readers to state that erdafitinib is an anti-FGFR therapy.
Response: The statement was added (p. 2).
Introduction
- Although this is well elaborated in the discussion section, this reviewer would emphasize the relevance of liquid biopsy, the advantages, significance, limitations, and value in resistance detection, in tumors that are rarely re-biopsied.
Response: The relevance of liquid biopsy was now emphasized in the introduction (p. 3).
Material and methods
- Since there is data from sequencing of cfDNA and tumor tissue material, this should be clarified before in the text for all ED because the interpretation of the results depends on that.
Response: The information specifying the source of the specimens in each of the analyzed databases was included (p. 4; the statement can be found in 1-3 paragraph describing each of the data-sources). - In the in-house TASMC cohort, authors provide some details on the NGS kit and genetic material employed. If possible, they should provide more details on the NGS protocol. Does the FusionFlex kit use DNA or RNA material? How was the quality assessed? These results might be important to show as supp data, because the genetic material from FFPE might compromise the results.
Response: The information was added according to your suggestion (p. 4).
- Some information on the PET-CT scans (which is in the discussion) should be included as material and methods.
Response: The information was added according to your suggestion (p. 4).
Results
- The first two sections might be fused in one as ‘FGFR2/3 fusion frequency in electronic databases.
Response: As you previously mentioned, GH and TASMC EDs differ significantly in terms of the NGS techniques, patient volume, baseline patient characteristics and clinical patient set-up. Therefore, we prefer not to combine these two databases but cite each of those separately.
- In the different paragraphs with the information from GH and TASMC EDs, difference on the origin of the sample (plasma and tumor tissue) should be clearly stated.
Response: The information specifying the source of the specimens in each of the analyzed databases was included (p. 4; the statement can be found in 1-3 paragraph describing each of the data-sources).
- Text in Figure 2 pie charts should be larger to be readable.
Response: The text in the figure was enlarged.
- This reviewer thinks that supp tables could be presented as main, and not supplementary.
Response: Supplementary Table S1&S2 were changed for tables 1&2 respectively.
- In Table s1, authors could add percentages to the number of cases. Although some percentages are displayed in the pie chart, this would more clearly associate these two interpretations of the numbers.
Response: Considering the very limited number of patients in most of the subgroups (1-4), and limited number of patients overall, we found more appropriate to present the numbers and not the percentiles.
- I would recommend using more widely used abbreviations, such as LUAD or ADC (or ADC-SCC), instead of adenoca.
Response: The abbreviations were incorporated according to your suggestion.
Discussion
- Discussion is very complete and clarifying.
Response: We are glad that you found the discussion complete and clarifying!
- Major: Regarding the statement, “We also acknowledge that most cases of FGFR fusions in GH ED occurred without a concomitant EGFR mutation, and therefore, in the majority of cases, FGFR fusion represents a driver by itself.”, Would you have information on other driver co-mutations in these patients? This would be relevant information to add even in one more figure if possible.
Response: The main focus of the manuscript was the FGFR fusions, so, unfortunately, we don't have this kind information.
- Little discussion on the occurrence of these fusions in patients not subsidiary of other approved therapies.
Response: It is difficult to estimate the exact prevalence of acquired FGFR fusion following treatment with EGFR TKIs in EGFR mutant aNSCLC because the testing is not mandatory. We expect that this will change in the coming 1-3 years, and our manuscript is one in the series supporting the concept.
We are looking forward to your favorable consideration.
Sincerely,
On the authors’ behalf,
Elizabeth Dudnik, MD
Head of the Lung Cancer Service, Assuta Medical Centers
Chair of the Israeli Lung Cancer Group, ISCORT
Ha Barzel St. 20, Tel-Aviv, 6971028
Phone: +972 3 7645177 +972 54 2678948
E-mail: elizabethd@assuta.co.il elizabeth.dudnik1603@gmail.com

Reviewer 2 Report
This research focused on acquired Resistance of EGFR-Mutant aNSCLC, which is an important problem for individual therapeutic strategy and prognosis prediction.
Dataset of a large number of aNSCLC patients treated with EGFR TKIs were tested by using liquid biopsy. Among which 164 Patients with FGFR fusions were analyzed, including 39 (23.8%) of cases co-existed with EGFR sensitizing mutations.
Firstly, how to get the clinical data such as age, sex, histologic types, and stage should be introduced and statistical method is also needed to describe the ordinal relationships of EGFR-Mutants. Secondly, clinical characteristics of the cohort should be analyzed in forms of words as well as a table.
Author Response
Dr. Emmanuel Andrès
Journal of Clinical Medicine
Editor-in-Chief
April 20, 2022
Re: “FGFR fusions as an acquired resistance mechanism following treatment with epidermal growth factor receptor tyrosine kinase inhibitors (EGFR TKIs) and a suggested novel target in advanced non-small cell lung cancer (aNSCLC)”
Dear Dr. Emmanuel Andrès,
Dear Reviewer,
On behalf of my co-authors, I would like to thank you for considering our revised manuscript for publication in Journal of Clinical Medicine. We are grateful for the helpful comments of the reviewers and for the editorial work. Considerable corrections have been made according to reviewers' comments – please, find below the list of points which were addressed.
Reviewer #2:
This research focused on acquired Resistance of EGFR-Mutant aNSCLC, which is an important problem for individual therapeutic strategy and prognosis prediction.
Dataset of a large number of aNSCLC patients treated with EGFR TKIs were tested by using liquid biopsy. Among which 164 Patients with FGFR fusions were analyzed, including 39 (23.8%) of cases co-existed with EGFR sensitizing mutations.
Firstly, how to get the clinical data such as age, sex, histologic types, and stage should be introduced and statistical method is also needed to describe the ordinal relationships of EGFR-Mutants.
Response: Thank you for your valuable comment! Regarding the GH ED, the available clinical data referring to the histological tumor subtype is provided in the Results section (p. 5) and Table 1. Unfortunately, the majority of the NGS databases (such as Foundation One, GH ED) lack the linking to the demographic and expanded clinical data (Flatiron is the only exception to that rule). We acknowledge this study limitation in the Discussion section (p. 10). We did provide, however, the clinical data for the 3 clinical cases, which, hopefully, allowed to illustrate the manuscript concept.
Secondly, clinical characteristics of the cohort should be analyzed in forms of words as well as a table.
Response: As previously mentioned, the clinical characteristics of the GH ED cohort can be found in the Results section (p. 5 and Talbe 1). The clinical characteristics of the case-series (3 patients) can be found in the Results section (pp. 8-9), and also summarized in Table 3.
We are looking forward to your favorable consideration.
Sincerely,
On the authors’ behalf,
Elizabeth Dudnik, MD
Head of the Lung Cancer Service, Assuta Medical Centers
Chair of the Israeli Lung Cancer Group, ISCORT
Ha Barzel St. 20, Tel-Aviv, 6971028
Phone: +972 3 7645177 +972 54 2678948
E-mail: elizabethd@assuta.co.il elizabeth.dudnik1603@gmail.com

Round 2
Reviewer 1 Report
Authors have addressed most of the points raised by this reviewer satisfactory. I would only add to this revision two comments on the responses to previous concerns:
- Major: Regarding the statement, “We also acknowledge that most cases of FGFR fusions in GH ED occurred without a concomitant EGFR mutation, and therefore, in the majority of cases, FGFR fusion represents a driver by itself.”, Would you have information on other driver co-mutations in these patients? This would be relevant information to add even in one more figure if possible.
Response: The main focus of the manuscript was the FGFR fusions, so, unfortunately, we don't have this kind information.
Reviewer#1: Without such information, I would re-phrase the sentence as “…FGFR fusion might represent a driver itself.”
- Little discussion on the occurrence of these fusions in patients not subsidiary of other approved therapies.
Response: It is difficult to estimate the exact prevalence of acquired FGFR fusion following treatment with EGFR TKIs in EGFR mutant aNSCLC because the testing is not mandatory. We expect that this will change in the coming 1-3 years, and our manuscript is one in the series supporting the concept.
Reviewer#1: Sorry that the comment was not elaborated enough. This reviewer thinks that, beyond the description of the FGFR fusions in the context of EGFR alteration and EGFR TKI resistance, few lines in the discussion could be dedicated to the current status of FGFR TKIs in lung cancer patients without any other actionable mutations (such as EGFR or others). This suggestion aims at providing a more complete view of the topic in a wider clinical setting.
Author Response
Dr. Emmanuel Andrès
Journal of Clinical Medicine
Editor-in-Chief
April 20, 2022
Re: “FGFR fusions as an acquired resistance mechanism following treatment with epidermal growth factor receptor tyrosine kinase inhibitors (EGFR TKIs) and a suggested novel target in advanced non-small cell lung cancer (aNSCLC)”
Dear Dr. Emmanuel Andrès,
Dear Reviewer,
On behalf of my co-authors, I would like to thank you for considering our revised manuscript for publication in Journal of Clinical Medicine. We are grateful for the helpful comments of the reviewers and for the editorial work. Considerable corrections have been made according to reviewers' comments – please, find below the list of points which were addressed.
Reviewer #1:
In this report titled: “FGFR fusions as an acquired resistance mechanism following treatment with epidermal growth factor receptor tyrosine kinase inhibitors and a suggested novel target in advanced non-small cell lung cancer”, Raphael et al. present an observational study in different and valuable advanced NSCLC patient databases. As others did before, they find a relatively low prevalence of FGFR fusions, and some of them occurring de novo upon anti-EGFR TKI treatment. For this, they take advantage of the Guardant Health electronic database, with information of liquid biopsy to detect FGFR fusions, together with other local databases. They also show real results in this clinical scenario in a small set of patients (n=2), in which patients developing resistance to anti-EGFR treatment, get considerably clinical benefit under clinically available anti-FGFR therapy. The paper is easy to read and clear, in general terms. While the concept is not really novel (authors cite some similar reports very adequately), this report adds information with potential implication to precision oncology of aNSCLC. Out of the numerous patients under anti-EGFR therapies, some will eventually acquire resistance through FGFR related mechanisms. The article points to two relevant aspects of the clinical scenario: the potential of liquid biopsy to monitor known resistance mechanisms and the possible tailored therapy for arising FGFR fusions. The percentage of patients with these fusions might seem small, but this is the basis of precision oncology.
Here I add some comments:
Abstract
- It would be good to mention the sample size for each cohort in the abstract (as they do for the last one n=3).
Response: Thank you for your valuable comment! Was added according to your suggestion (p. 2).
- It would also help to non-expert readers to state that erdafitinib is an anti-FGFR therapy.
Response: The statement was added (p. 2).
Introduction
- Although this is well elaborated in the discussion section, this reviewer would emphasize the relevance of liquid biopsy, the advantages, significance, limitations, and value in resistance detection, in tumors that are rarely re-biopsied.
Response: The relevance of liquid biopsy was now emphasized in the introduction (p. 3).
Material and methods
- Since there is data from sequencing of cfDNA and tumor tissue material, this should be clarified before in the text for all ED because the interpretation of the results depends on that.
Response: The information specifying the source of the specimens in each of the analyzed databases was included (p. 4; the statement can be found in 1-3 paragraph describing each of the data-sources). - In the in-house TASMC cohort, authors provide some details on the NGS kit and genetic material employed. If possible, they should provide more details on the NGS protocol. Does the FusionFlex kit use DNA or RNA material? How was the quality assessed? These results might be important to show as supp data, because the genetic material from FFPE might compromise the results.
Response: The information was added according to your suggestion (p. 4).
- Some information on the PET-CT scans (which is in the discussion) should be included as material and methods.
Response: The information was added according to your suggestion (p. 4).
Results
- The first two sections might be fused in one as ‘FGFR2/3 fusion frequency in electronic databases.
Response: As you previously mentioned, GH and TASMC EDs differ significantly in terms of the NGS techniques, patient volume, baseline patient characteristics and clinical patient set-up. Therefore, we prefer not to combine these two databases but cite each of those separately.
- In the different paragraphs with the information from GH and TASMC EDs, difference on the origin of the sample (plasma and tumor tissue) should be clearly stated.
Response: The information specifying the source of the specimens in each of the analyzed databases was included (p. 4; the statement can be found in 1-3 paragraph describing each of the data-sources).
- Text in Figure 2 pie charts should be larger to be readable.
Response: The text in the figure was enlarged.
- This reviewer thinks that supp tables could be presented as main, and not supplementary.
Response: Supplementary Table S1&S2 were changed for tables 1&2 respectively.
- In Table s1, authors could add percentages to the number of cases. Although some percentages are displayed in the pie chart, this would more clearly associate these two interpretations of the numbers.
Response: Considering the very limited number of patients in most of the subgroups (1-4), and limited number of patients overall, we found more appropriate to present the numbers and not the percentiles.
- I would recommend using more widely used abbreviations, such as LUAD or ADC (or ADC-SCC), instead of adenoca.
Response: The abbreviations were incorporated according to your suggestion.
Discussion
- Discussion is very complete and clarifying.
Response: We are glad that you found the discussion complete and clarifying!
- Major: Regarding the statement, “We also acknowledge that most cases of FGFR fusions in GH ED occurred without a concomitant EGFR mutation, and therefore, in the majority of cases, FGFR fusion represents a driver by itself.”, Would you have information on other driver co-mutations in these patients? This would be relevant information to add even in one more figure if possible.
Response: The main focus of the manuscript was the FGFR fusions, so, unfortunately, we don't have this kind information.
Reviewer#1: Without such information, I would re-phrase the sentence as “…FGFR fusion might represent a driver itself.”
Response: The sentence was re-phrased according to your suggestion.
- Little discussion on the occurrence of these fusions in patients not subsidiary of other approved therapies.
Response: It is difficult to estimate the exact prevalence of acquired FGFR fusion following treatment with EGFR TKIs in EGFR mutant aNSCLC because the testing is not mandatory. We expect that this will change in the coming 1-3 years, and our manuscript is one in the series supporting the concept.
Reviewer#1: Sorry that the comment was not elaborated enough. This reviewer thinks that, beyond the description of the FGFR fusions in the context of EGFR alteration and EGFR TKI resistance, few lines in the discussion could be dedicated to the current status of FGFR TKIs in lung cancer patients without any other actionable mutations (such as EGFR or others). This suggestion aims at providing a more complete view of the topic in a wider clinical setting.
Response: The discussion was expanded, and now includes brief discussion of the current status of FGFR TKIs in lung cancer patients with FGFR aberrations.
We are looking forward to your favorable consideration.
Sincerely,
On the authors’ behalf,
Elizabeth Dudnik, MD
Head of the Lung Cancer Service, Assuta Medical Centers
Chair of the Israeli Lung Cancer Group, ISCORT
Ha Barzel St. 20, Tel-Aviv, 6971028
Phone: +972 3 7645177 +972 54 2678948
E-mail: elizabethd@assuta.co.il elizabeth.dudnik1603@gmail.com
This manuscript is a resubmission of an earlier submission. The following is a list of the peer review reports and author responses from that submission.